# Generalizable Robotic Manipulation: Object-Centric Diffusion Policy with Language Guidance

Hang Li[*,1,2], Qian Feng[*,1,2], Zhi Zheng[1], Jianxiang Feng[1,2], Alois Knoll[1],

*Abstract*—**Learning from demonstrations faces challenges in generalizing beyond the training data and is fragile even to slight visual variations. To tackle this problem, we introduce Lan-o3dp, a language-guided object-centric diffusion policy that takes 3d representation of task-relevant objects as conditional input and can be guided by cost function for safety constraints at inference time. Lan-o3dp enables strong generalization in various aspects, such as background changes, camera view shift and visual ambiguity, and can avoid novel obstacles that are unseen during the demonstration process. Specifically, We first train a diffusion policy conditioned on point clouds of target objects and then harness a large language model to decompose the user instruction into task-related units consisting of target objects and obstacles, which can be used as visual observation for the policy network or converted to a cost function, guiding the generation of trajectory towards collision free region at test time. Our proposed method shows training efficiency and higher success rates compared with the baselines in simulation experiments. In real-world experiments, our method exhibits strong generalization performance towards unseen instances, cluttered scenes, scenes of multiple similar objects and demonstrates training-free capability of obstacle avoidance.**

## I. INTRODUCTION

Recently, diffusion models have shown great potential in the field of robotic manipulation [17, 25, 31]. In the realm of imitation learning, diffusion based methods [10, 42] have demonstrated strong capabilities towards learning complex manipulation tasks. Compared to traditional imitation learning algorithms, diffusion models offer the advantages of stable training, high-dimensional output spaces, and the ability to capture the multi-modal distribution of actions [10].

However, their performance is limited when testing scenes are different from training in terms of scenes of similar objects but with different appearances, new backgrounds, camera view shift, unseen obstacles, and so on. Generalization across different scenarios and safe deployment are crucial for the widespread application of robotics. To address the challenges of generalization and safe obstacle avoidance, we propose Lan-o3dp, a language-guided, object-centric collision-aware diffusion policy. By leveraging large language model and vision language model, our approach identifies task-relevant objects in the scene based on language instructions, segments the relevant objects point clouds from the overall scene point cloud, and encodes this data into a compact 3D representation. The diffusion model is trained to predict robot end effector

trajectory conditioned on these 3D representations of task-relevant objects by denoising random noise into a coherent action sequence. Moreover, at testing stage, it can also segment the point cloud of obstacles in addition to target objects based on the language instruction. The location and geometric information of obstacles are obtained from the segmented point cloud and can be used to construct a cost function. The gradient of the cost function is then integrated into guided sampling [13] to direct the trajectory prediction phase, allowing the robotic arm's end effector to avoid obstacles. The training free obstacle avoidance can be effectively integrated into the proposed 3D representation pipeline, as it allows for the acquisition of the location and geometric information of obstacles from a calibrated camera on one hand, and on the other hand, the policy does not fail due to the changes in the scene caused by the addition of extra obstacles.

To evaluate our method, we conduct experiments across seven simulated tasks in the RLBench [16] environment and three real-world tasks. We demonstrate the effectiveness of our proposed methods against state-of-the-art diffusion-based methods in simulation and further evaluate the generalization and obstacle avoidance capabilities of our method in the real world.

In summary, our contributions are three-fold:

1) We propose Lan-o3dp, an effective language-guided collision-aware visuomotor policy that generalizes across diverse aspects such as background changes, camera view shift, and even scenes of multiple similar objects.
2) We introduce a novel guidance mechanism to avoid obstacles by analyzing the limitations of the existing guided sampling approaches. We present theoretical explanations and validations on real robots. Given language instructions, Lan-o3dp identifies novel obstacles and avoids collision given training data of no obstacles, which further improves the generalization capability.
3) The proposed method is evaluated in both simulation and real world experiments and shows the effectiveness and universality compared to baselines.

## II. METHOD

### A. Problem Formulation

In this work, we address the generalization problem of the diffusion policy and introduce collision awareness at inference time by adjusting the visual conditioning and sampling guidance.

*: Equal Contributions, {hang1.li, qian.feng}@tum.de
[1]Technical University of Munich
[2]Agile Robot

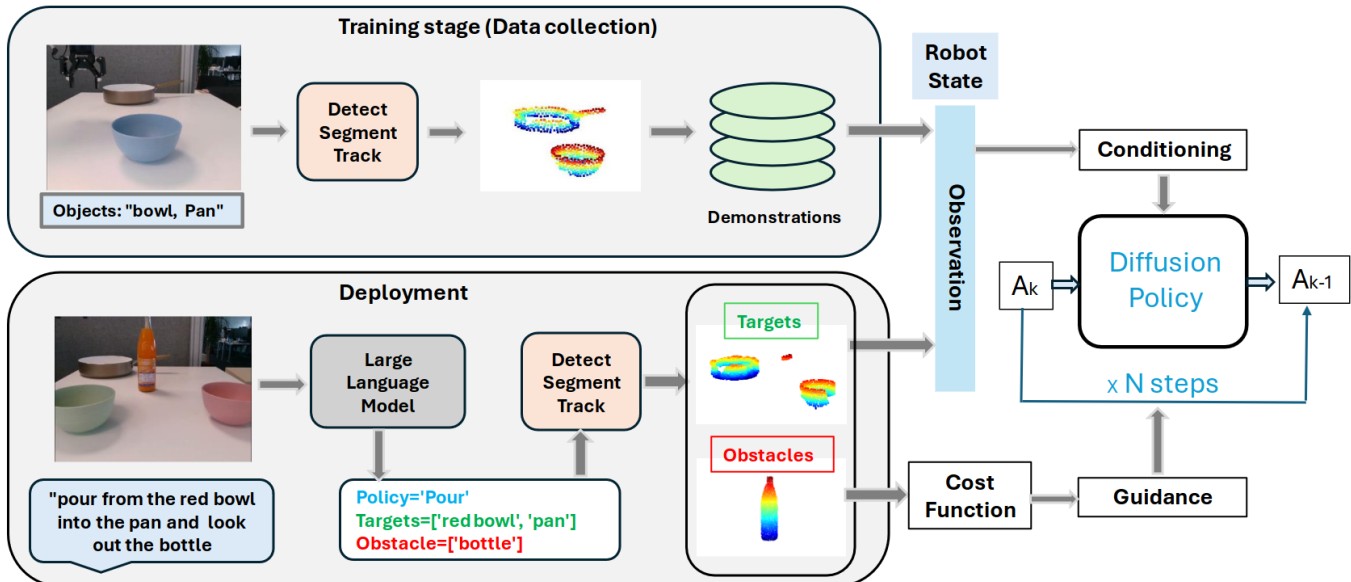

Fig. 1: An illustration of the proposed pipeline of Lan-o3dp. At the training stage, the visual observations in the demonstrations we collected only contained objects relevant to the task. During the deployment phase, we utilize a large language model to decompose users' instructions into target objects and obstacles and select the corresponding policy. Target objects are used as visual observation for the model, while obstacles are transformed into a cost function to guide the model in generating collision-free trajectories.

**Diffusion policy visual conditioning:** Diffusion policy[10] uses DDPM to model the action sequence $P(A_t \mid O_t)$. Wherein, $A_t = \{at, \dots, at+n\}$ is the predicted next $n$ action steps, which is a sequence of end-effector poses. The prediction horizon $n$ indicates that the diffusion policy predicts a trajectory over a shorter horizon instead of the entire trajectory. $O_t = \{V_t, S_t\}$ represents the visual observation $V_t$ and robot states observation $S_t$. The observation features are fused to the policy network through Film [30]. While diffusion policy takes pure RGB images as visual observation, we employ the segmented point clouds of task-relevant objects from a calibrated camera for policy learning. By removing redundant visual information and retaining only task-relevant information, our model can minimize the negative effects of scene changes, thereby improving generalization performance.

**Guided sampling formulation:** The diffusion model is trained to predict the added noise $\epsilon(O, A_k, k)$ at each diffusion timestep $k$, and during the reverse diffusion process, it gradually denoises a Gaussian noise to a smooth noise-free trajectory. The reverse process step is $A_{k-1} = \mu_k + \sigma_k z$, where $\mu_k = \frac{1}{\sqrt{\alpha_k}} \left( A_k - \frac{1-\alpha_k}{\sqrt{1-\overline{\alpha}_k}} \varepsilon \right)$, $z \sim \mathcal{N}(0, I)$, $\alpha_k \in \mathbb{R}$ and $\overline{\alpha}_k := \prod_{s=1}^{k} \alpha_s$ predefined scheduling parameters. The subscript time $t$ of the trajectory index is dropped for ease of notation. Much prior work has explored guided sampling of the diffusion model. At the inference stage, guidance $g_k = \nabla_{A_k} D$ as a gradient term of cost/distance $D$ with respect to $A_k$ is added to the model's predicted mean such that each denoising step becomes:

$$A_{k-1} = \mu_k - \rho g_k + \sigma_k z \qquad (1)$$

, where $\rho$ is a scaling factor to control the effect of guidance. In this work, we model newly emerged obstacles in the scene

as a cost function to guide the model in generating collision-free trajectories. The cost gradient and distance gradient have opposite signs for avoiding obstacles. Benefiting from the object-centric 3D representation of the pipeline, we can obtain the locations and basic geometric information of obstacles within the scene.

*B. Approach*

**Training stage:** To obtain task-relevant target point clouds, we leverage open vocabulary segmentation to acquire real-time masks of the target objects and map these masks onto the point clouds. Figure 1 shows our pipeline. As shown in the training stage, before starting to record the demonstrations, the task related objects are specified simply by words. A vision language model (VLM) is firstly called to detect corresponding objects within the scene to obtain the bounding boxes of the target objects, which are then passed as prompt to Segment Anything Model (SAM) [18] to obtain segmentation masks. Upon completion of segmentation, recording commences, and a video object segmentation model is employed to track the objects in real-time. The tracked masks are projected to point clouds, resulting in point cloud representations of the objects for each frame. The point clouds of objects are further downsampled by farthest point sampling.

**Language guided deployment:** During the deployment phase, our model is applicable to different scenarios. We use a large language model to decompose the user's commands into policy, target objects, and obstacles. Similarly, open vocabulary segmentation is used to obtain the point cloud of the target objects and obstacles in each frame. The point cloud of target objects is subsequently inputted as an observation into the trained policy, while the point cloud of obstacles is

processed and transformed into a cost function. The gradient of this cost function is then utilized to guide the trajectory generation towards collision free areas.

**Cost guided generation:** In the field of robotics, many guided sampling techniques rely on reward models [17, 21], which are, however, often difficult to obtain. We choose to use a flexibly constructed cost function instead. To convert obstacles information to cost function, we calculate the distance between every waypoint in the generated action sequence and the centers of obstacles $C_{ob}$. As previously mentioned, most guided sampling methods calculate the cost/distance $D(A_k, C_{ob})$ of each intermediate action $A_k$ generated during the reverse diffusion process and compute the gradient $g_k = \nabla_{A_k} D(A_k, C_{ob})$. However, a cost function that is independent of the timestep $k$ of the diffusion process becomes less meaningful because of the noisy trajectories, especially in the early stages of the denoising process. Consequently, the cost of noisy trajectories struggles to provide effective guidance. Unlike previous methods [32, 6], refer to FreeDoM [40], we calculate the cost at each step based on the estimated $A_{0|k}$, an estimated clean trajectory.

$$A_{0|k} := \mathbb{E}[A_0 | A_k] = \frac{A_k - \sqrt{1 - \overline{\alpha}_k}\epsilon_\theta(A_k)}{\sqrt{\overline{\alpha}_k}} \quad [14] \quad (2)$$

We calculate the cost/distance of $A_0$ estimated from $A_k$ at each timestep and use this cost to compute the gradient with respect to $A_k$, that is $\nabla_{A_k} D(A_{0|k}, C_{ob})$. Therefore, the equation 1 becomes:

$$A_{k-1} = \mu_k - \rho\nabla_{A_k} D(A_{0|k}, C_{ob}) + \sigma_k z \quad (3)$$

As discussed in [40], it is difficult to achieve effective guidance during the early stages of the diffusion process due to too chaotic sample. We choose to guide the generation during specific time periods. The detailed algorithm is shown in **Algo 1**

---

**Algorithm 1** Cost guided diffusion sampling, given a diffusion model $\epsilon_\theta$, cost/distance measurement $D(x, y)$, and gradient scale $\rho$.

---

1: $A_T \leftarrow$ sample from $\mathcal{N}(0, I)$
2: **for** $k = T$ to $1$ **do**
3:  $\mu_k \leftarrow \frac{1}{\sqrt{\alpha_k}}\left(A_k - \frac{1-\alpha_k}{\sqrt{1-\overline{\alpha}_k}}\epsilon_\theta\right)$
4:  $A_{k-1} \leftarrow \mu_k + \sigma_k z$
5:  **if** $k \leq S$ **then**
6:    $A_{0|k} = \frac{A_k - \sqrt{1-\overline{\alpha}_k}\epsilon_\theta(A_k)}{\sqrt{\overline{\alpha}_k}}$
7:    $A_{k-1} = A_{k-1} - \rho\nabla_{A_k} D(A_{0|k}, C_{ob})$
8:  **end if**
9: **end for**
10: **Return** $A_0$

---

## III. EXPERIMENTS

In our experiments, we show the following aspects: (1) Object-centric 3d diffusion policy achieves a higher average

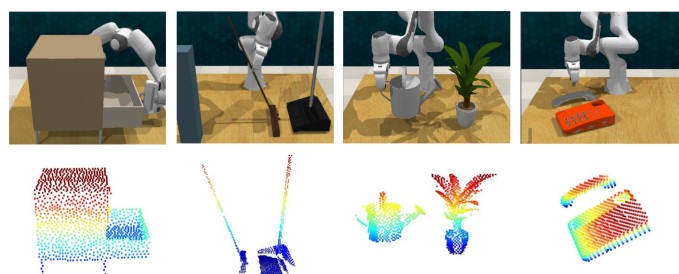

Fig. 2: Visualization of some simulation tasks. We use a single front camera and extract the segmented point cloud of task-related objects to keep them consistent with the real world. The top row shows RGB images from the front camera, and the bottom row shows visualizations of the corresponding object point clouds.

success rate in simulation experiments compared to the baselines; (2) Our method has strong generalization capabilities for scene changes; and (3) Cost guided generation can effectively avoid language specified obstacles.

### A. Simulation Experiments

We conduct simulation experiments in RLBench to evaluate the success rate of proposed Lan-o3dp compared with two baselines namely diffusion policy [10] and 3D diffusion policy [42]. To keep consistent with real-world experiments, we only use the front camera and collect 40 demonstrations for each task across 7 tasks, which cover manipulation, pick-and-place, single object, and multiple objects. Examples are shown in figure 2, we extract the task related object point cloud. Each demonstration of every task has variation, such as position changes of the objects.

We use the convolutional network based diffusion policy. We train 500 epochs for each task, evaluate 20 episodes every 50 epochs, and then compute the average of the highest 5 success rates. The episodes for evaluation also have variations. As shown in table I achieves an overall 68.8 % success rate across 7 RLBench tasks.

### B. Real world Experiments

In the real-world experiments, we aim to verify the generalization in the following aspects: (1) instance changes, (2) multiple similar objects, (3) camera view shift, and (4) language informed obstacle avoidance.

*1) Experiment Setup:*

*a)* **System setup and task design**: We conduct real-world experiments on 3 tasks with a Diana 7 robot arm. We use one RealSense D415 camera to capture the RGB image and point cloud. Our tasks are Bowl_pour: grasp the bowl and pour the contents of the bowl into the pan; Bottle_upright: stand the horizontal bottle upright. Bottle_in_drawer: put the bottle into the drawer. We use GPT-4 [29] at testing time to extract policy, target objects and obstacles from user instruction and generate code to run the policy.

*b)* **Demonstrations collection**: Demonstrations are collected by teleoperation with a space mouse and keyboard. We collect 40 demonstrations for each task and the training scenes

TABLE I: Simulation Results on Rlbench

| Tasks | open drawer | open wine bottle | sweep to dustpan | phone on base | put item in drawer | water plants | close microwave | Average Succ Rate |
|---|---|---|---|---|---|---|---|---|
| Diffusion Policy | 70.0% | 38.0% | 57.0% | 11.0% | 32.0% | **41.0%** | **95.0%** | 49.2% |
| 3D Diffusion | **94.0%** | 49.0% | 66.0% | 6.0% | 5.0% | 21.0% | 94.0% | 49.6% |
| Lan-o3dp | 90.0% | **77.0%** | **77.0%** | **57.0%** | **50.0%** | 37.5% | 93.0% | **68.8%** |

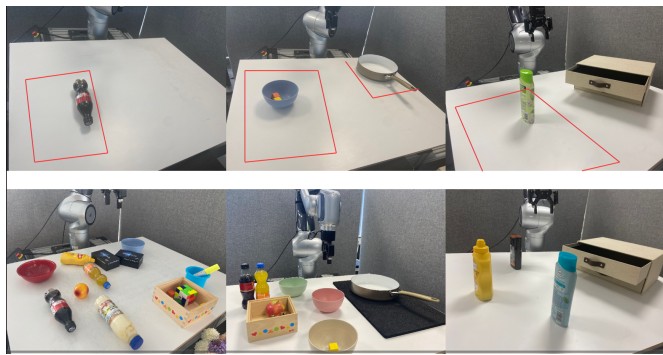

Fig. 3: Training scenes (top row) and cluttered test scenes (bottom row). The red lines indicate the position variations in collected demonstrations. From left to right, the tasks are bottle upright, bowl pour, and bottle in drawer.

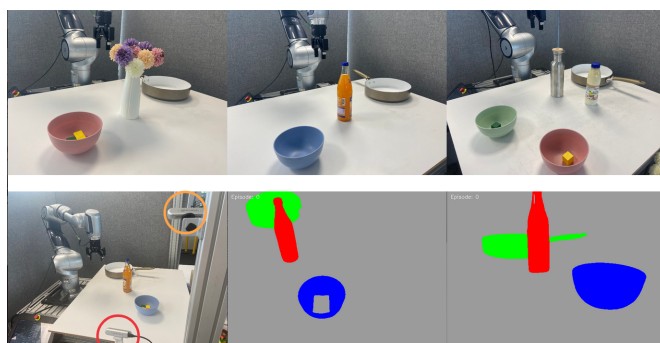

Fig. 4: Scenes with obstacles (top row) and camera view shift (bottom row). The top row is the task bowl pour with novel obstacles. The bottom row is the setup of two cameras and the segmentation masks of the current scene from two cameras.

and position variation are shown in the upper part of figure 3. In the bottle upright task and bottle in drawer task, the orientation of the bottle is not changed. Given task-related objects, we invoke open vocabulary detector GroundingDINO [23] to predict the bounding boxes, Segment Anything [18] to obtain the segmentation masks, and finally track the masks using video tracker Cutie [8]. We record observations consisting of objects point clouds and state observations, including the robot end effector poses and gripper state.

*2) Generalization Evaluation:*

*a) Instance changes:* We evaluate the generalization ability to objects with similar geometry through all three tasks. We have observed that our model can handle changes in the appearance of objects, but it tends to perform less effectively with objects that undergo larger changes in geometric shape.

*b) Scenes of multiple similar objects:* In scenes with multiple objects (figure 3), language models are crucial due to the visual ambiguity caused by the presence of multiple similar objects. We use natural language to specify which object should be the target.

*c) Camera view shift:* As shown in figure 4 bottom row, we use the camera in the red circle for demonstration collection and the camera in the orange circle is only for testing. We observed that there is no performance drop when the camera changes from red circle to orange circle.

*3) Testing time obstacles avoidance:*

*a) Scenes of obstacles:* We test the cost guided obstacle avoidance in the bowl pouring task shown in figure 4. We construct the cost and let the generated trajectory change horizontally. The obstacles are modeled as cylinders, and the radius is the sum of the diameter of the bowl and the radius of the obstacle in the horizontal orientation. We notice that the robot can successfully avoid obstacles and finish the tasks. The gradient scale significantly influences the quality of the

generated trajectory and the effect of obstacle avoidance.

*b) Cost function constructed from obstacles:* We calculate the distance to the center of obstacles $C_{ob}$ for all waypoints $a_i$ in the estimated trajectory $A_{0|k}$. If any distance $D(a_i, C_{ob})$ in $D(A_{0|k}, C_{ob})$ is shorter than a safety critical distance $Q^*$, a non-zero gradient will be assigned to the corresponding waypoint.

$$Gradient = \begin{cases} \nabla D(a_i, C_{ob}), & \text{if } D(a_i, C_{ob}) \leq Q^* \\ 0, & \text{if } D(a_i, C_{ob}) > Q^* \end{cases} \quad (4)$$

In real world experiments, we only consider the distance of x and y coordinates.

## IV. CONCLUSION

In this work, we use the point cloud of the target objects as the input for the diffusion policy model to enhance the model's generalization performance. By filtering out objects that are irrelevant to the task, our model can perform well in changed scenes. Additionally, we have introduced training free cost-guided trajectory generation for obstacle avoidance, converting necessary obstacles into costs to achieve safer deployment. This work has some limitations. We utilize the point cloud of objects as the output of the model, assuming that the target object can be successfully detected by the model. The performance of the current VLM for detecting objects is limited, which restricts the performance of our model. Additionally, our modeling of obstacles uses a relatively simple distance measure, which is insufficient for complex obstacles. Future work could involve using more advanced visual-language models, incorporating a task planner for complex task planning, and modeling obstacles in more detail to avoid more complex obstacles.

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

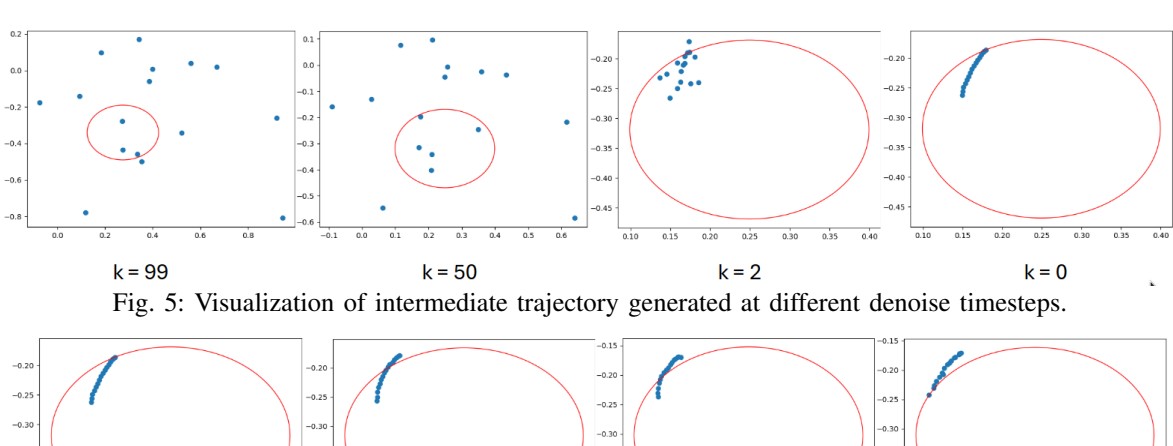

k = 99     k = 50     k = 2     k = 0

Fig. 5: Visualization of intermediate trajectory generated at different denoise timesteps.

No guidance     $\rho = 0.0001$     $\rho = 0.0002$     $\rho = 0.0003$

Fig. 6: Effect of different gradient scale.

## V. APPENDIX

### A. Related Work

*a) Diffusion Models in Robotics:* Diffusion model is a type of probabilistic generative model that learns to generate new data by progressively applying a denoising process on a randomly sampled noise. Learning based robotic grasping [19, 26, 5] and manipulation skills [43, 22] are longstanding problems. Due to its advantage of stable training and impressive expressiveness, diffusion model has been applied in several robotic fields such as reinforcement learning [2, 17, 21], imitation learning [10, 42], grasp synthesis [3, 38] and motion planning [36, 6, 32]. In this work, we utilize object-centric 3D representation to maximize the generalization ability of the trained policy based on imitation learning via diffusion model and introduce a novel guided diffusion mechanism for obstacle avoidance.

*b) Object-Centric Representation Learning:* Object-centric representations have been widely studied to reason about visual observations modularly in the robotic field. In robotics, researchers commonly use 6D poses [34, 35, 27], bounding boxes [37, 12] or segmented masks [11] to represent objects in a scene. These representations are limited to known object categories or instances. Recent progress in open-world visual recognition has led to the development of substantial models across various domains, including object detection [23], object segmentation [18], video object segmentation [7]. Groot[43] trains a transformer policy using segmented 3D objects. However, Groot uses a Segmentation Correspondence Model to identify the target object and cannot handle scenes with multiple similar objects. We use open vocabulary segmentation, which allows for a more convenient specification of target objects through language.

*c) Language models for robotics:* Large language models(LLMs) possess powerful language comprehension abilities and a wealth of common knowledge. As a result, they can be effectively utilized to understand human instructions and to plan robotic tasks at a high level. Code as Policies[20] explored using a code writing LLM to generate robot policy code based on language commands. Voxposer[15] plans the task and generates code by a LLM to compose value maps for zero shot manipulation. SayCan[1] use a LLM to select skills from a library of pre-trained skills. Many works also explore using LLMs to write reward functions for training robotic skills [41, 39, 24]. In addition, With the help of pre-trained open vocabulary vision language model [23, 28, 9], the robot can ground the user's instruction to the real world and accomplish various and complex tasks [15, 33, 4]. In this work, we use a language model to select the desired policy and extract objects and obstacles from the user's instructions and obtain an object point cloud with the vision language model.

### B. Details of real world experiment

*a) Visualization and explanation of proposed guidance mechanism:* The generated trajectory of the denoising process is too chaotic especially at the early stage as illustrated in figure 5. The cost of trajectory with too much noise is barely meaningful. We calculate the cost of estimated clean trajectory $A_{0|k}$ instead. The red circle in the following figures indicates the obstacle and the blue points are the waypoints of the generated action sequence. The visualization is only about x and y coordinations.

The impact of different gradient scales $\rho$ on the generated final trajectory is shown in figure 6. A point within the red circle is a point has collision with the obstacle. In our experiments, a gradient scale greater than 0.0003 can effectively avoid obstacles.

*b) Gripper state for policy learning:* Since the proposed method is robot agnostic, the policy does not know whether the robot has successfully grasped the target object. We add a simple feedback signal from the gripper (1 indicates the object was grasped, while 0 means nothing in gripper) as additional robot state observation to the policy and greatly improve the success rate of all tasks.