# OpenReview forum: "Generalizable Robotic Manipulation: Object-Centric Diffusion Policy with Language Guidance"
_roboticsfoundation.org/RSS/2024/Workshop/EARL — EARL 2024 Poster_

### Official Review · Reviewer_oPPB · 2024-06-24
**A nice framework for leveraging open vocabulary segmentation for obtaining obstacle information to incorporate into diffusion denoising**

**Rating:** 7
**Confidence:** 4

**Review:**

The authors propose a pipeline that builds upon recent developments in Vision Language Models (VLMs) and Diffusion Policies for generating collision-free actions via Imitation Learning. Given a set of demonstrations, the authors leverage Vision-Language Models to detect, segment, and track the 3D pointclouds of relevant objects in the scene. These segmented demonstrations are then used to train a Diffusion Policy for the task at hand. During testing, the denoising step in the diffusion process is performed by using Guided Sampling wherein the gradient of a cost function is additionally added in the denoising for a cost-aware diffusion process. The authors use the segmented pointcloud to calculate the collision cost of the denoised trajectory at the current denoising step(Eq. 2), which is then used in the guided sampling for denoising (Eq. 3). The authors show that their approach achieves better success rates than a standard diffusion policy. The paper is well written and has a nice contribution both from a theoretic and an engineering standpoint, therefore it is a clear accept.

Minor Comments:
- The time subscripts in the description $A_t$ in Sec. III.A should be fixed. Additionally it would be helpful to mention that the subscript for time is dropped for ease of notation.
- It is not clear what the actions are. Is it a full trajectory that is considered like in [4] and [24] or is it just the next immediate end-effector/joint positions?
- I'm assuming Eq. 2 is paraphrased from Eq. 15 in [12]. If so, then the denominator $\sqrt{\alpha_k}$ should apply for $A_k$ as well and the equation would look like $(A_k - \sqrt{1-\alpha_k}\epsilon_\theta(A_k))/\sqrt{\alpha_k}$

---

### Decision · Program_Chairs · 2024-06-24

Accept (Poster)